# Antifungal Activity of Methylxanthines against Grapevine Trunk Diseases

Eva Sánchez-Hernández [1], Celia Andrés-Juan [2], Laura Buzón-Durán [1], Adriana Correa-Guimaraes [1], Jesús Martín-Gil [1,*] and Pablo Martín-Ramos [3,*]

1   Agricultural and Forestry Engineering Department, ETSIIAA, Universidad de Valladolid, Avenida de Madrid 44, 34004 Palencia, Spain; eva.sanchez.hernandez@uva.es (E.S.-H.); laura.buzon.duran@gmail.com (L.B.-D.); adriana.correa@uva.es (A.C.-G.)

2   Department of Organic Chemistry, Facultad de Ciencias, Universidad de Valladolid, Calle Paseo de Belén 7, 47011 Valladolid, Spain; celia.andres.juan@uva.es

3   Instituto Universitario de Investigación en Ciencias Ambientales de Aragón (IUCA), EPS, Universidad de Zaragoza, Carretera de Cuarte s/n, 22071 Huesca, Spain

\*   Correspondence: mgil@iaf.uva.es (J.M.-G.); pmr@unizar.es (P.M.-R.)

**Abstract:** Methylxanthines, found in the seeds, leaves, and fruits of some plants, are receiving increasing attention as promising treatments for wood-degrading fungi. The aim of the study presented herein was to explore the potential applications of caffeine, four caffeine derivatives (viz. 8-bromo-caffeine, 8-iodo-caffeine, 8-(4-fluorophenoxy)-caffeine, and 8-(2,3,5,6-tetrafluoroalcoxy)-caffeine), and theophylline as antifungals for *Botryosphaeriaceae* species associated with grapevine trunk diseases (GTDs). In vitro susceptibility tests were conducted to assess the antimycotic activity of the aforementioned compounds and their conjugated complexes with chitosan oligomers (COS). Caffeine, Br-caffeine, and I-caffeine exhibited higher efficacies than imidazole, the chosen antifungal control. Moreover, a strong synergistic behavior between COS and the methylxanthine derivatives was observed. The COS–I-caffeine complex showed the best overall performance against the phytopathogenic fungi with $EC_{90}$ values of 471, 640, and 935 μg mL$^{-1}$ for *D. seriata*, *D. viticola*, and *N. parvum*, respectively. In a second step, combinations of the new treatments with imidazole were also explored, resulting in further activity enhancement and $EC_{90}$ values of 425, 271, and 509 mL$^{-1}$ against *D. seriata*, *D. viticola*, and *N. parvum*, respectively, for the COS–I-caffeine-imidazole ternary compound. Given the high in vitro efficacy of these formulations for the control of GTDs, they may deserve further investigation with in vivo and field bioassays as an alternative to conventional fungicides.

**Keywords:** antifungals; caffeine; *Diplodia seriata*; *Dothiorella viticola*; imidazole; *Neofusicoccum parvum*; chitosan; theophylline

## 1. Introduction

Caffeine (1,3,7-trimethylxanthine or 1,3,7-trimethyl-3,7-dihydro-1H-purine-2,6-dione) is an alkaloid of the xanthine group produced from the coffee bean but also found in the fruits, nuts, seeds or leaves of more than 80 plant species native to South America, East Asia, and Africa. It is related to the guanine and adenine bases of DNA and RNA. It can stimulate the central nervous system by antagonizing the adenosine receptors on neurons, and thus, temporarily stimulates users. It is the most widely used psychoactive drug in the world. Theophylline (1,3-dimethylxanthine) is another naturally occurring xanthine derivative, present in tea and cocoa. It is also a central nervous stimulant and is widely used as a bronchodilator in the treatment of asthma and chronic obstructive pulmonary disease.

Beyond their versatile pharmaceutical applications, natural and synthetic methylxanthines inhibit insect feeding and are pesticidal at concentrations known to occur in plants, and, at lower concentrations, they are potent synergists for other pesticides [1].

Concerning the antimicrobial activity of these substances, the antibacterial activity of caffeine is modest-low and it has been necessary to synthesize and test 8-substituted-aryl-alkoxy derivatives to improve it [2]. On the other hand, methylxanthines feature an interesting antifungal activity, given that they inhibit fungal chitinases that are necessary for fungal cell wall remodeling and cell replication [3]. In fact, several studies have put forward caffeine—and more recently, theophylline—as ecological, safe, and affordable alternatives to conventional biocides for wood protection [4–7].

In view of the aforementioned activity against brown-rot and white-rot fungi, other applications of methylxanthines as antifungal products may be envisaged. For instance, they could be particularly useful against fungi that cause grapevine trunk diseases (GTDs), a major challenge faced by modern viticulture. While a variety of inorganic and organic compounds, either synthetic or natural, have been investigated for the prevention and control of these phytopathologies in the past two decades [8], to the best of our knowledge, methylxanthines have not been tested on GTDs.

To fill this research gap, the aim of the study presented herein was to conduct a screening of the activity of the natural methylxanthines, caffeine, and theophylline, together with four synthetic caffeine derivatives (Figure 1), against three fungi that cause GTDs. In addition, the aforementioned methylxanthines were tested in combination with chitosan oligomers (COS), which also possess antimicrobial properties [9] and have been successfully used for entrapping methylxanthines for pharmaceutical applications [10–13], to optimize their activity, bioavailability, and gradual release. As a third and partial goal of this research, possible synergies resulting from combinations of COS–methylxanthine complexes with imidazole were explored since benzimidazoles (benomyl, carbendazim, methyl thiophanate, etc.) are among the most effective agents against GTDs, but they cannot protect pruning wounds during the whole period in which they are susceptible to GTDs (ranging from 2 to 4 months) [8].

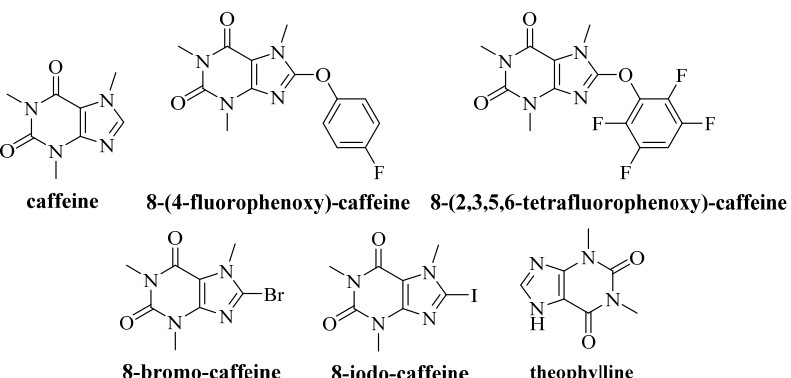

**Figure 1.** Methylxanthines tested.

## 2. Material and Methods

### 2.1. Chemical Reagents

Chitosan (CAS 9012-76-4; high molecular weight: 310,000–375,000 Da) was obtained from Hangzhou Simit Chem. & Tech. Co. (Hangzhou, China). Neutrase™ 0.8 L enzyme was provided by Novozymes A/S (Bagsværd, Denmark). Caffeine (CAS 58-08-2), 8-bromo-caffeine (CAS 10381-82-5), 8-iodo-caffeine (CAS 5415-41-8), 2,3,5,6-tetrafluorophenol (CAS 769-39-1), theophylline (CAS 58-55-9), imidazole (CAS 288-32-4), and methanol (UHPLC, CAS 67-56-1) were purchased from Sigma-Aldrich Química (Madrid, Spain). Potato dextrose agar (PDA) was provided by Becton Dickinson (Bergen County, NJ, USA).

Chitosan oligomers (COS) were prepared according to the procedure published in [14] with the modifications detailed in [15] to obtain a solution with a pH of 4–6 with oligomers having a molecular weight less than 2000 Da and a polydispersity index of 1.6.

### 2.2. Synthesis of 8-(4-fluorophenoxy)-caffeine and 8-(2,3,5,6-tetrafluorophenoxy)-caffeine

The synthesis of the two caffeine derivatives was carried out according to the procedure outlined in Figure 2. In a round bottom flask with a reflux condenser, 1.36 g (0.005 mol) of 8-bromo-caffeine was mixed with 0.84 g (0.0057 mol) of 4-fluorophenol or 1.245 g (0.0075 moles) of 2,3,5,6-tetrafluorophenol, 0.69 g (0.005 mol) of anhydrous $K_2CO_3$ and 150 mL of dry N,N-dimethylformamide (DMF). The mixture was heated at 100 °C for 8 h under a $N_2$ atmosphere. After heating was complete, the mixture was cooled to room temperature, water was added and the resulting reaction mixture was kept at 4 °C until a white solid appeared which, in both cases, was filtered and recrystallized in hexane.

**Figure 2.** Schematic of the synthesis of 8-(4-fluorophenoxy)-caffeine and 8-(2,3,5,6-tetrafluorophenoxy)-caffeine.

### 2.3. Preparation of Conjugate Complexes

COS–caffeine/theophylline/caffeine derivative conjugates were obtained by mixing a solution of COS with caffeine, theophylline, and caffeine derivatives in a 1:1 (*w/w*) ratio, followed by sonication for fifteen minutes with five three-minute pulses to prevent the temperature from exceeding 60 °C. A 1000 W, 20 kHz UIP1000hdT probe-type ultrasonicator (Hielscher Ultrasonics, Teltow, Germany) was used. The formation of the conjugated complexes was confirmed by infrared spectroscopy. Imidazole was added to these conjugates in an appropriate ratio to form ternary 1:1:1 COS–caffeine derivatives-imidazole complexes that were processed, like the binary complexes, by sonication.

### 2.4. Characterization of Caffeine Derivatives by Nuclear Magnetic Resonance Spectroscopy

$^1$H NMR (400 or 500 MHz) and $^{13}$C NMR (100 or 126 MHz) spectra were recorded in CDCl$_3$. Proton chemical shifts were reported in ppm relative to tetramethylsilane using the residual CHCl$_3$ resonance as an internal reference. The chemical shifts for carbon were reported in ppm relative to tetramethylsilane and refer to solvent carbon resonance. Data were reported as follows: chemical shift, multiplicity (s = singlet, d = doublet, t = triplet, q = quartet, m = multiplet, and br = width), coupling constants in Hz, and integration.

### 2.5. Physico-Chemical Characterization of Conjugated Complexes

A Nicolet iS50 Fourier-transform infrared spectrometer (Thermo Scientific, Waltham, MA, USA) equipped with a diamond attenuated total reflection (ATR) system was used to collect the infrared vibrational spectra over the 400–4000 cm$^{-1}$ range with a spectral resolution of 1 cm$^{-1}$. Interferograms resulting from the co-addition of 64 scans were used.

Thermogravimetric (TG) and differential scanning calorimetry (DSC) analyses were carried out using a TG-DSC2 apparatus (Mettler Toledo; Columbus, OH, USA) in a $N_2$:$O_2$ (4:1) atmosphere with a heating ramp of 20 °C min$^{-1}$.

### 2.6. Energy Content and Molecular Structure of Imidazole Complexes with Caffeine Derivatives

Energy calculations were performed with ChemBio3D Ultra 16.0 software (Perkin-Elmer Inc., Waltham, MA, USA) and the Molecular Mechanics (MM2) calculation method.

### 2.7. Fungal Isolates

The following fungal isolates were provided as PDA subcultures by ITACYL (Valladolid, Spain) and used for in vitro tests: *Dothiorella viticola* A.J.L. Phillips & J. Luque (code ITACYL_F118, isolate Y-103-08-01), *Diplodia seriata* De Not. (code ITACYL_F098, isolate Y-084-01-01-01a), and *Neofusicoccum parvum* (Pennycook & Samuels) Crous, Slippers & A.J.L. Phillips (code ITACYL_F111, isolate Y-091-03-01c).

### 2.8. In Vitro Antimicrobial Activity Assessment

The antifungal activity of each treatment was determined using the agar dilution method [16] and incorporating aliquots of stock solutions into the PDA medium to provide final concentrations in the 62.5–1500 $\mu$g mL$^{-1}$ range. Mycelial plugs with a 5 mm diameter of *D. seriata*, *D. viticola,* and *N. parvum* from the margin of seven-day-old PDA cultures were transferred to plates filled with the amended media (three plates per treatment and concentration combination; each experiment was carried out twice). Plates containing only PDA were used as a control. Radial mycelium growth was determined by calculating the average of two perpendicular colony diameters for each replicate. Mycelium growth inhibition was calculated after seven days of incubation in the dark at 25 $^{\circ}$C according to the formula: $((d_c - d_t)/d_c) \times 100$, where $d_c$ is the average fungal colony diameter in the control and $d_t$ is the average diameter of the treated fungal colony. The 50% and 90% effective concentrations (EC$_{50}$ and EC$_{90}$, respectively) were estimated using PROBIT analysis in IBM SPSS Statistics v.25 software (IBM; Armonk, NY, USA). Wadley's method [17] was chosen to determine the level of interaction, i.e., the synergy factors (SFs).

### 2.9. Statistical Analysis

The results of mycelial growth inhibition for *D. seriata*, *D. viticola*, and *N. parvum* by the different concentrations tested for each treatment were statistically analyzed in IBM SPSS Statistics v.25 software using a two-way analysis of variance, followed by Tukey's test at $p < 0.05$ for the *post hoc* comparison of means (given that homogeneity and homoscedasticity requirements were fulfilled according to the Shapiro–Wilk and Levene tests).

## 3. Results

### 3.1. Characterization of Caffeine Derivatives by Nuclear Magnetic Resonance Spectroscopy

1,3,7-trimethyl-3,7-dihydro-1H-purine-2,6-dione. $^{1}$H NMR (500 MHz, CDCl$_3$): $\delta$ = 3.39 (s, 3H), 3.57 (s, 3H), 3.98 (s, 3H), 7.51 (s, 1H). $^{13}$C NMR (126 MHz, CDCl$_3$): $\delta$ = 27.9, 29.7, 33.5, 107.5, 141.3, 148.5, 151.6, 155.3.

8-(4-fluorophenoxy)-1,3,7-trimethyl-3,7-dihydro-1H-purine-2,6-dione. $^{1}$H NMR (500 MHz, CDCl$_3$): $\delta$ = 3.39 (s, 3H), 3.43 (s, 3H), 3.87 (s, 3H), 7.10 (m, 2H), 7.27 (m, 2H). $^{13}$C NMR (126 MHz, CDCl$_3$): $\delta$ = 27.8, 29.8, 30.3, 103.8, 116.2, 116.4, 121.1, 121.2, 145.7, 149.0 (d, J = 2.9 Hz), 151.6, 153.3, 154.9, 159.0, 161.0. $^{19}$F NMR (470 MHz, CDCl$_3$): $\delta$ = −116.6.

1,3,7-trimethyl-8-(2,3,5,6-tetrafluorophenoxy)-3,7-dihydro-1H-purine-2,6-dione. $^{1}$H NMR (500 MHz, CDCl$_3$): $\delta$ = 3.38 (s, 6H), 3.94 (s, 3H), 7.09 (tt, J = 7.0, 9.8 Hz, 1H). $^{13}$C NMR (126 MHz, CDCl$_3$): $\delta$ = 27.8, 29.8, 30.6, 103.6, 103.8, 104.0, 104.5, 140.7 (dd, J = 253.2, 15.3 Hz), 145.3, 146.4 (m), 151.4, 152.0, 154.9. $^{19}$F NMR (470 MHz, CDCl$_3$): $\delta$ = −137.9 (m, 2F), −153.4 (m, 2F).

8-bromo-1,3,7-trimethyl-3,7-dihydro-1H-purine-2,6-dione. $^{1}$H NMR (500 MHz, CDCl$_3$): $\delta$ = 3.38 (s, 3H), 3.54 (s, 3H), 3.95 (s, 3H). $^{13}$C NMR (126 MHz, CDCl$_3$): $\delta$ = 28.0, 29.8, 33.9, 109.3, 128.1, 148.0, 151.2, 154.4.

8-iodo-1,3,7-trimethyl-3,7-dihydro-1H-purine-2,6-dione. $^{1}$H NMR (500 MHz, CDCl$_3$): $\delta$ = 3.39 (s, 3H), 3.55 (s, 3H), 3.94 (s, 3H). $^{13}$C NMR (126 MHz, CDCl$_3$): $\delta$ = 28.0, 29.8, 36.1, 101.0, 110.7, 149.5, 151.3, 154.1.

### 3.2. Vibrational Characterization

#### 3.2.1. COS–Caffeine Derivative Conjugates

The FTIR spectrum of caffeine showed the two characteristic peaks of methyl C–H group vibrations (2950–2850 cm$^{-1}$). In the 1600–1400 cm$^{-1}$ range (Table 1), C=N and C=C bond vibrations [18] were observed, and in the 1330–1000 cm$^{-1}$ interval, the in-plane bending vibrations of C–H, C–N, and C–C bonds were recorded. The peaks at 1200 cm$^{-1}$ fell into the spectral region for the tensile vibrations of the >C=O ketone carbonyl group and C–N bonds.

The COS spectrum (not tabulated) showed absorption at 3400 cm$^{-1}$ (–OH stretching), 2880 cm$^{-1}$ (–CH stretching), 1650 cm$^{-1}$ (–NH$_2$ bending), 1600 cm$^{-1}$ (–NH$_2$ bending), 1324 cm$^{-1}$ (C–N bending), and 565 cm$^{-1}$ (pyranoside ring).

Conjugation of the COS–caffeine and COS–caffeine derivatives was evidenced by the shift of the caffeine bands in the 1690–1480 cm$^{-1}$ region to higher wavenumbers: 1693→1695, 1644→1652, 1546→1548, 1480→1481 cm$^{-1}$ for caffeine→COS–caffeine; and 1695→1697 and 1652→1659 for 8-iodo-caffeine→COS–8-iodo-caffeine (in this latter case, a new band appeared at 1553 cm$^{-1}$). An interaction of the NH$^{3+}$ groups in COS with active groups in caffeine/caffeine derivatives was indicated by a shift of the NH$_2$ bending vibration from 1650 cm$^{-1}$ in COS and 1644 cm$^{-1}$ in caffeine to 1652 cm$^{-1}$, 1656 cm$^{-1}$, and 1659 cm$^{-1}$ in COS–caffeine, COS–8-bromo-caffeine, and COS–8-iodo-caffeine, respectively. There was also a shift in the peak related to C–N bending at 1324 cm$^{-1}$ in COS and 1326 cm$^{-1}$ in caffeine to 1335–1343 cm$^{-1}$ in the COS–caffeine derivatives.

In the lower wavenumber region, complexation with COS induced the appearance of new bands that, in the case of COS–caffeine, were located at 846, 723, 715, 672, 589, 538, and 503 cm$^{-1}$. In contrast, the absorption band at 644 cm$^{-1}$ for non-complexed caffeine disappeared. In the case of COS–8-iodo-caffeine, apart from the bands present in 8-iodo-caffeine, absorption bands appeared at 724, 715, 617, 549, and 446 cm$^{-1}$.

The FTIR spectrum of theophylline (not tabulated) included bands at 1705, 1661, 1563, 1483, 1441, 1315, 1284, 1239, 1186, 1047, 978, 948, 926, 913, and 846 cm$^{-1}$. The FTIR spectrum of COS–theophylline-imidazole (not tabulated) showed bands at 1714, 1667, 1560, 1442, 1378, 1321, 1241, 1187, 1089, 1064, 1024, 980, 891, 834, 763, 742, 660, 629, 610, 556, 502, 447, and 446 cm$^{-1}$.

#### 3.2.2. COS–Caffeine Derivative-Imidazole Complexes

The formation of ternary complexes by imidazole incorporation into the COS–caffeine derivative conjugates led to small, but appreciable, band shifts in the 1720–1650 cm$^{-1}$ region (between 2 and 3 cm$^{-1}$ towards higher wavenumbers, Table 2). Such shifts are compatible with an increase in the degree of aggregation by π-stacking formation since the intermolecular overlap of π-stacking orbitals leads to exciton delocalization.

### 3.3. Thermal Analysis of Conjugated Compounds

From the TG curves of the COS–caffeine derivatives conjugated compounds (Table 3), three characteristic weight loss temperature intervals were observed. The first one (25–350 °C), which produced 50–70% weight loss (except for COS–iodo-caffeine, where there was 10% weight loss), was created by a thermal effect between 200 and 250 °C, corresponding to the decomposition of the caffeine moiety (caffeine decomposes completely at about 285 °C), and a thermal effect at 330–340 °C, corresponding to the depolymerization of the chitosan oligomer chains. The second weight loss was less than 43% (except for COS–iodo-caffeine, where it was 53%), occurred at 400–560 °C, and was caused by the decomposition of the COS polymeric chains by deacetylation and cleavage of glycosidic bonds. The last stage, at a temperature above 560 °C, resulted in 2–19% weight loss, except for the COS–iodo-caffeine complex (37%). Since the COS–iodo-caffeine complex only suffered a weight loss of 10% at temperatures up to 500 °C, it can be concluded that this conjugate is the most stable in this series.

**Table 1.** Absorption bands of caffeine, its conjugate with chitosan oligomers (COS), and the conjugates of its derivatives (fluorophenoxy-caffeine, tetrafluorophenoxy-caffeine, bromo-caffeine, and iodo-caffeine) with COS. Wavenumbers of the bands are expressed in $cm^{-1}$.

| Caffeine | COS–Caffeine | COS–Fluorophenoxy-Caffeine | COS–Tetrafluorophenoxy-Caffeine | COS–Bromo-Caffeine | COS–Iodo-Caffeine | Assignment |
|---|---|---|---|---|---|---|
| 1693 | 1708 | 1701 | 1706 | 1701 | 1697 | stretching vibration of conjugated C=O(2) and C=O(6) carbonyl groups |
| 1644 | 1652 | 1646 | 1659 | 1656 | 1659 | C=N and C=C stretching in imidazoles and $NH_2$ bending |
| 1598 | 1598 | | 1614 | 1605 | | C=N vibration and $NH_2$ bending |
| 1546 | 1548 1520 | 1557 1525 | 1556 1519 | 1562 | 1557 1538 | C=N and C=C vibrations |
| 1480 | 1481 | 1504 | 1492 | | | C–N vibration |
| 1455 | 1456 | 1473 1441 | 1455 | 1449 | 1441 | associated with methyl groups |
| 1429 1398 | 1429 | | 1403 | 1404 | | C=C and C–H stretching and C=N deformation |
| 1358 | 1360 | 1380 | | | 1380 | C–H deformation |
| 1326 | | | 1335 | 1341 | 1342 1321 | C−N stretching |
| 1285 | 1285 1272 | 1286 | 1284 1273 | 1283 | 1276 | C=C and C=O stretching |
| 1239 | 1238 | 1245 1200 | 1215 | 1252 1215 | 1256 1214 | stretching of the >C=O ketonic carbonyl group and C−N bonds |
| 1189 1133 | 1185 1137 | 1152 | 1185 1151 1136 | 1185 1151 1136 | 1185 1151 | C–N vibration in imidazoles and C–O stretching of the alcoholic hydroxyl group in chitosan |
| 1067 | 1067 | 1069 | 1068 | 1067 | 1067 | CH out-of-plane deformation |
| 1024 | 1027 | 1024 | 1031 | 1033 | 1031 | C–C stretching; C–N stretching; >C=O (ketonic) group |
| 972 | 977 946 | 970 | 964 946 | 972 | 974 946 | combination of $C–CH_3$ vibrations and C–H deformation |
| | 846 | 894 807 | 864 847 | 800 | 895 836 | C–C stretching and/or out-of-plane hydrogen deformation of the NH group |

**Table 2.** Absorption bands in the 1700–1650 cm$^{-1}$ wavenumber range for the conjugates of caffeine derivatives (fluorophenoxy-caffeine, tetrafluorophenoxy-caffeine, bromo-caffeine, and iodo-caffeine) with chitosan oligomers (COS) and their corresponding ternary complexes with imidazole.

| COS–Caffeine | COS–Caffeine-Imidazole | COS–F-Phenoxy Caffeine | COS–F-Phenoxy Caffeine-Imidazole | COS–4F-Phenoxycaffeine | COS–4F-Phenoxycaffeine-Imidazole | COS–Bromo-Caffeine | COS–Bromo-Caffeine-Imidazole | COS–Iodo-Caffeine | COS–Iodo-Caffeine-Imidazole |
|---|---|---|---|---|---|---|---|---|---|
| 1695 | 1698 | 1699 | 1701 | 1710 | 1712 | 1702 | 1705 | 1697 | 1698 |
| 1652 | 1659 | 1646 | 1648 | 1659 | 1663 | 1656 | 1668 | 1659 | 1661 |

**Table 3.** Thermal effects (°C) and weight loss (%) along the degradation stages observed in the TG curves for the COS conjugates with caffeine and its derivatives.

| Conjugate Complex | 1st Stage | Weight Loss | 2nd Stage | Weight Loss | 3rd Stage | Weight Loss |
|---|---|---|---|---|---|---|
| COS–caffeine | 230, 340 | 70% | 465, 525 | 28% | >560 | 2% |
| COS–Fluorophenoxy-caffeine | 248, 345 | 54% | 480, 558 | 29% | >560 | 16% |
| COS -Tetrafluorofluorophenoxy-caffeine | 248, 337 | 63% | 500, 560 | 18% | >560 | 19% |
| COS–Bromo-caffeine | 205, 330 | 49% | 455, 533 | 42% | >560 | 8% |
| COS–Iodo-caffeine | 225 | 10% | 497, 515 | 53% | >560 | 37% |

Melting points of reference products: caffeine, 235–237 °C; bromo-caffeine, 209 °C; iodo-caffeine, 260–264 °C.

### 3.4. Energy Content and Molecular Structure of Imidazole Interaction with Caffeine Derivatives

Table 4 shows the energy content calculated for the complexes of caffeine and caffeine derivatives with imidazole. It should be noted that the lower the energy content, the higher the stability of the complex. Thus, according to MM2 calculations, the least stable complex would be iodo-caffeine-imidazole. The 5-linked nitrogen heterocycles of imidazole and the caffeine derivatives would be stabilized when the two rings are arranged in parallel due to π-stacking interactions.

**Table 4.** Energy content of the conjugates of caffeine and its derivatives (fluorophenoxy-caffeine, tetrafluorophenoxy-caffeine, bromo-caffeine, and iodo-caffeine) with imidazole, chitosan oligomers (COS) with imidazole, and the COS–caffeine-imidazole complex.

| Conjugate Complex | Energy (kcal mol$^{-1}$) |
| --- | --- |
| Caffeine-imidazole | 45.72 |
| Caffeine-imidazole (parallel rings) | 41.24 |
| Fluorophenoxy-caffeine-imidazole | 47.78 |
| Tetrafluorophenoxy-caffeine-imidazole | 55.95 |
| Bromo-caffeine-imidazole | 43.88 |
| Iodo-caffeine-imidazole | 56.06 |
| COS–imidazole | 132.43 |
| COS–caffeine-imidazole | 197.46 |

Regarding the ternary COS–iodo-caffeine-imidazole complex (Figure 3), modeling allowed us to observe that the iodo-caffeine-imidazole interaction was maintained by π-stacking.

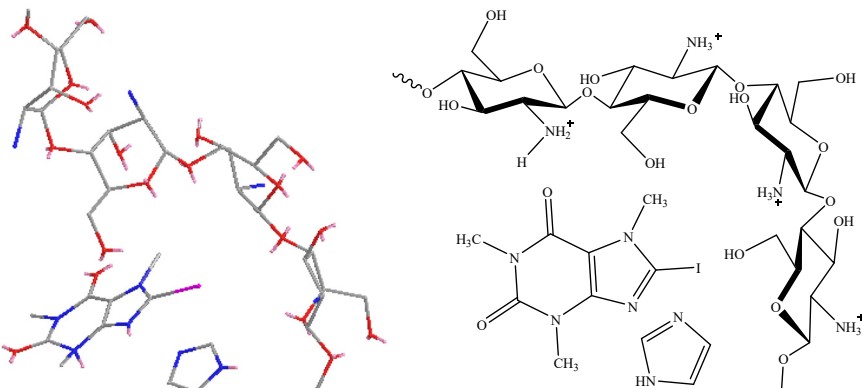

**Figure 3.** Structural modeling of the COS–iodo-caffeine-imidazole complex.

### 3.5. In Vitro Antimicrobial Activity

The results of the antifungal susceptibility tests conducted for *D. seriata* (Figures S1 and S4), *D. viticola* (Figures S2 and S5), and *N. parvum* (Figures S3 and S6) are summarized in Figures 4–6, respectively. The lettered groupings for the treatment–concentration combinations according to Tukey's HSD test after a two-way ANOVA for each of the phytopathogens are presented in Table S1.

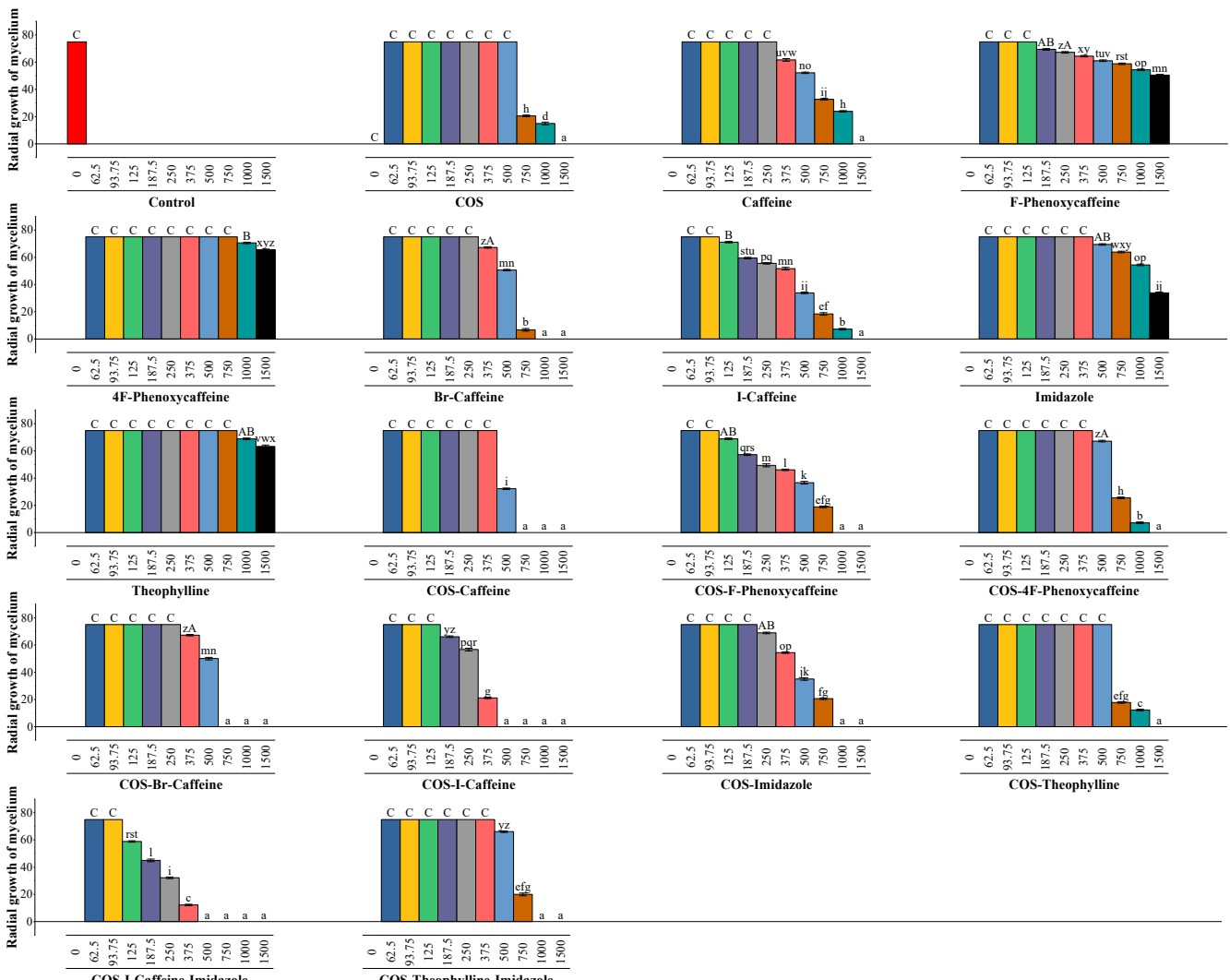

**Figure 4.** Mycelial growth inhibition of *D. seriata* by caffeine, its derivatives, imidazole, and theophylline, their respective COS–conjugated complexes, and some selected ternary combinations with imidazole at different concentrations (ranging from 62.5 to 1500 µg mL$^{-1}$). Error bars represent standard deviation. Treatment–concentration combinations accompanied by the same letters are not significantly different at $p < 0.05$ according to Tukey's HSD test after a two-way ANOVA (Tukey's HSD lettered grouping follows the sequence a–z, A–C, where 'a' indicates the group of treatment–concentration combinations with the highest efficacy and 'C' the group of least effective treatment–concentration combinations).

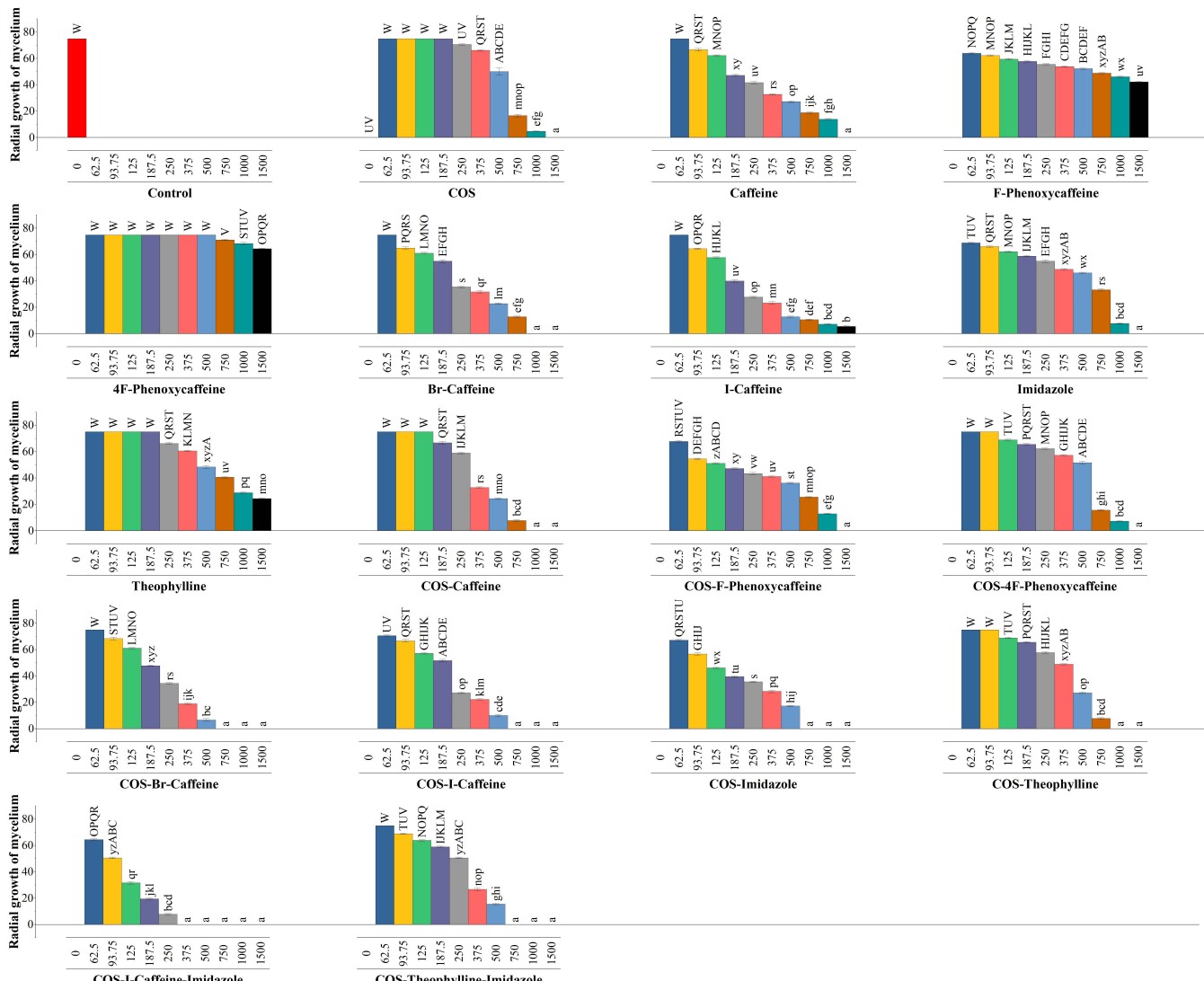

**Figure 5.** Mycelial growth inhibition of *D. viticola* by caffeine, its derivatives, imidazole, and theophylline, their respective COS–conjugated complexes, and some selected ternary combinations with imidazole at different concentrations (ranging from 62.5 to 1500 µg mL$^{-1}$). Error bars represent standard deviation. Treatment–concentration combinations accompanied by the same letters are not significantly different at $p < 0.05$ according to Tukey's HSD test after a two-way ANOVA (Tukey's HSD lettered grouping follows the sequence a–z, A–W, where 'a' indicates the group of treatment–concentration combinations with the highest efficacy and 'W' the group of least effective treatment–concentration combinations).

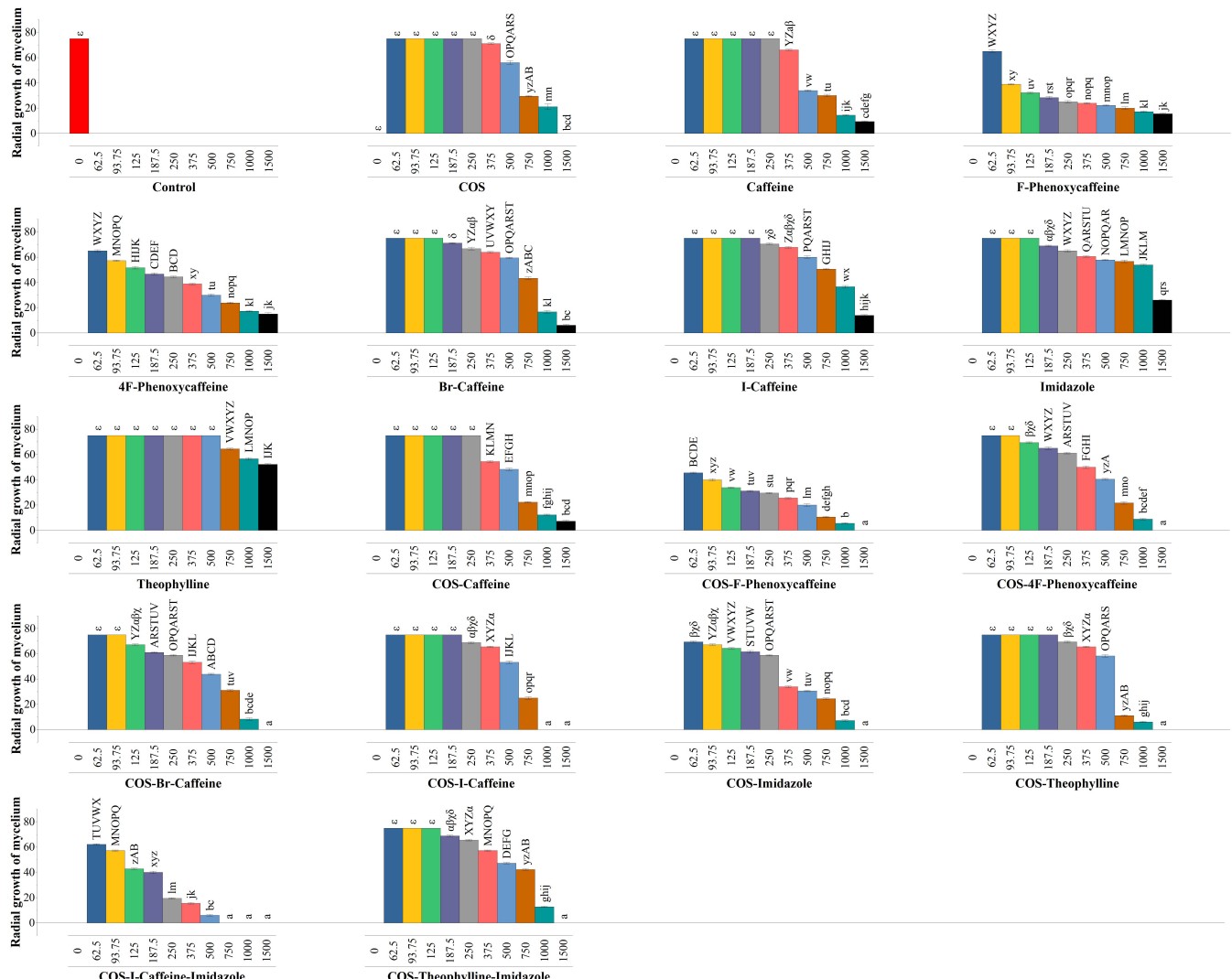

**Figure 6.** Mycelial growth inhibition of *N. parvum* by caffeine, its derivatives, imidazole, and theophylline, their respective COS–conjugated complexes, and some selected ternary combinations with imidazole at different concentrations (ranging from 62.5 to 1500 μg mL$^{-1}$). Error bars represent standard deviation. Treatment–concentration combinations accompanied by the same letters are not significantly different at $p < 0.05$ according to Tukey's HSD test after a two-way ANOVA (Tukey's HSD lettered grouping follows the sequence a–z, A–Z, α–ε, where 'a' indicates the group of treatment–concentration combinations with the highest efficacy and 'ε' the group of least effective treatment–concentration combinations).

If the above results are expressed as effective concentrations (Table 5), it can be seen that caffeine had an EC$_{90}$ of around 1300 μg mL$^{-1}$ against the three tested phytopathogens. Among its derivatives, bromo-caffeine and iodo-caffeine showed better antifungal behavior than caffeine against *D. seriata* and *D. viticola*, whereas fluorophenoxy-caffeine and tetrafluorophenoxy-caffeine showed very poor antifungal behavior (except against *N. parvum* in the case of fluorophenoxy-caffeine). Theophylline showed lower activity than caffeine, bromo-caffeine, and iodo-caffeine.

**Table 5.** Effective concentration values for chitosan oligomers (COS), caffeine, its derivatives, imidazole, and theophylline against three fungi of the *Botryosphaeriaceae* family.

| Treatment | Effective Concentration ($\mu$g mL$^{-1}$) | *D. seriata* | *D. viticola* | *N. parvum* |
|---|---|---|---|---|
| COS | EC$_{50}$ | 744 | 554 | 680 |
| | EC$_{90}$ | 1180 | 1139 | 1327 |
| Caffeine | EC$_{50}$ | 678 | 303 | 479 |
| | EC$_{90}$ | 1341 | 1352 | 1360 |
| Fluorophenoxy-caffeine | EC$_{50}$ | 684 | 2548 | 147 |
| | EC$_{90}$ | 7106 * | 67,913 * | 1863 |
| Tetrafluorophenoxy-caffeine | EC$_{50}$ | 11,786 * | 24,746 * | 362 |
| | EC$_{90}$ | 108,037 * | 561,164 * | 1985 |
| Bromo-caffeine | EC$_{50}$ | 554 | 292 | 769 |
| | EC$_{90}$ | 767 | 829 | 1379 |
| Iodo-caffeine | EC$_{50}$ | 492 | 237 | 1044 |
| | EC$_{90}$ | 968 | 753 | 2792 |
| Imidazole | EC$_{50}$ | 1407 | 611 | 1241 |
| | EC$_{90}$ | 2793 | 1190 | 2392 |
| Theophylline | EC$_{50}$ | 6947 | 820 | 2771 |
| | EC$_{90}$ | 42,213 * | 2689 | 11,454 * |

* Unreliable value due to difficulty with fitting the dose-response relationship in cases of low activity in the range of concentrations tested.

Complexation with COS resulted in enhanced antifungal activity in all cases (Table 6). Among the conjugate complexes, the COS–iodo-caffeine complex showed good efficacy against *D. seriata* and *N. parvum* with EC$_{90}$ values of 471 and 935 $\mu$g mL$^{-1}$, respectively; the COS–bromo-caffeine complex proved to be the most effective treatment for *D. viticola* with an EC$_{90}$ of 490 $\mu$g mL$^{-1}$; and the COS–fluorophenoxy-caffeine complex showed appreciable inhibition of *N. parvum* with an EC$_{90}$ value of 998 $\mu$g mL$^{-1}$ that was comparable to that of the COS–iodo-caffeine complex. The COS–theophylline conjugate complex showed a lower efficacy than COS–iodo-caffeine and COS–bromo-caffeine against *D. seriata* and *D. viticola*, but was more effective against *N. parvum* with an EC$_{90}$ value of 917 $\mu$g mL$^{-1}$. The values of the synergy factor (SF) calculated according to Wadley's method were greater than 1 for all COS complexes (Table 7), indicating synergistic behavior.

**Table 6.** Effective concentration values for the COS binary and ternary conjugate complexes against the three grapevine phytopathogens.

| Treatment | Effective Concentration ($\mu$g mL$^{-1}$) | *D. seriata* | *D. viticola* | *N. parvum* |
|---|---|---|---|---|
| COS–caffeine | EC$_{50}$ | 483 | 331 | 598 |
| | EC$_{90}$ | 727 | 789 | 1267 |
| COS–Fluorophenoxy-caffeine | EC$_{50}$ | 472 | 494 | 121 |
| | EC$_{90}$ | 900 | 1205 | 998 |
| COS–Tetrafluorofluorophenoxy-caffeine | EC$_{50}$ | 681 | 565 | 517 |
| | EC$_{90}$ | 943 | 960 | 1045 |
| COS–Bromo-caffeine | EC$_{50}$ | 539 | 238 | 571 |
| | EC$_{90}$ | 688 | 490 | 1192 |
| COS–Iodo-caffeine | EC$_{50}$ | 310 | 240 | 599 |
| | EC$_{90}$ | 471 | 640 | 935 |

**Table 6.** *Cont.*

| Treatment | Effective Concentration ($\mu g\ mL^{-1}$) | *D. seriata* | *D. viticola* | *N. parvum* |
|---|---|---|---|---|
| COS–Imidazole | $EC_{50}$ | 494 | 206 | 419 |
| | $EC_{90}$ | 910 | 695 | 1052 |
| COS–Theophylline | $EC_{50}$ | 729 | 439 | 593 |
| | $EC_{90}$ | 1155 | 804 | 917 |
| COS–Iodo-caffeine-imidazole | $EC_{50}$ | 215 | 116 | 163 |
| | $EC_{90}$ | 425 | 267 | 509 |
| COS–Theophylline-imidazole | $EC_{50}$ | 659 | 329 | 752 |
| | $EC_{90}$ | 880 | 705 | 1250 |

**Table 7.** Synergy factors for $EC_{90}$ concentrations (according to Wadley's method) for conjugated complexes of chitosan oligomers with caffeine-derivatives, imidazole, and theophylline.

| Pathogen | COS–Caffeine | COS–Br-caffeine | COS–I-caffeine | COS–F-phenoxy-caffeine | COS–4F-phenoxy-caffeine | COS–Imidazole | COS–Theophylline |
|---|---|---|---|---|---|---|---|
| *D. seriata* | 1.73 | 1.35 | 2.26 | 2.25 | 2.48 | 1.82 | 1.99 |
| *D. viticola* | 1.56 | 1.96 | 1.42 | 1.86 | 2.37 | 1.68 | 1.99 |
| *N. parvum* | 1.05 | 1.13 | 1.92 | 1.55 | 1.52 | 1.62 | 2.59 |

Regarding the ternary complexes (Figure 7, Table 6), the binary complex COS–iodo-caffeine was selected to form the ternary complex with imidazole because it had the best activity against two out of the three fungi (and the second-best inhibition value against the third). Even though the $EC_{90}$ values of imidazole alone were worse (higher) than those of COS and iodo-caffeine against *D. seriata* and *N. parvum*, and comparable in the case of *D. viticola*, the $EC_{90}$ values for the COS–imidazole complex were comparable with COS–iodo-caffeine (except against *D. seriata*), and the ternary mixture led to the lowest values among all tested compounds ($EC_{90}$ = 425, 267, and 509 $\mu g\ mL^{-1}$ against *D. seriata*, *D. viticola*, and *N. parvum*, respectively). In the case of the COS–theophylline-imidazole ternary complex, its activity was higher than that of the COS–theophylline binary complex against *D. seriata* and *D. viticola* (but not *N. parvum*), and its performance was worse than COS–iodo-caffeine-imidazole.

From the statistical comparison of all assayed treatments, if only the most effective treatment–concentration combinations (i.e., those in the 'a' group according to Tukey's HSD test) are selected (Table 8), it may be inferred that COS–I-caffeine-imidazole will be the most effective treatment. However, if imidazolic compounds are to be avoided due to legal restrictions (it is worth noting that, apart from sodium arsenite and methyl bromide, benzimidazole fungicides have also been phased out in some countries due to environmental and public health concerns [19]), then COS–I-caffeine would be the preferred choice.

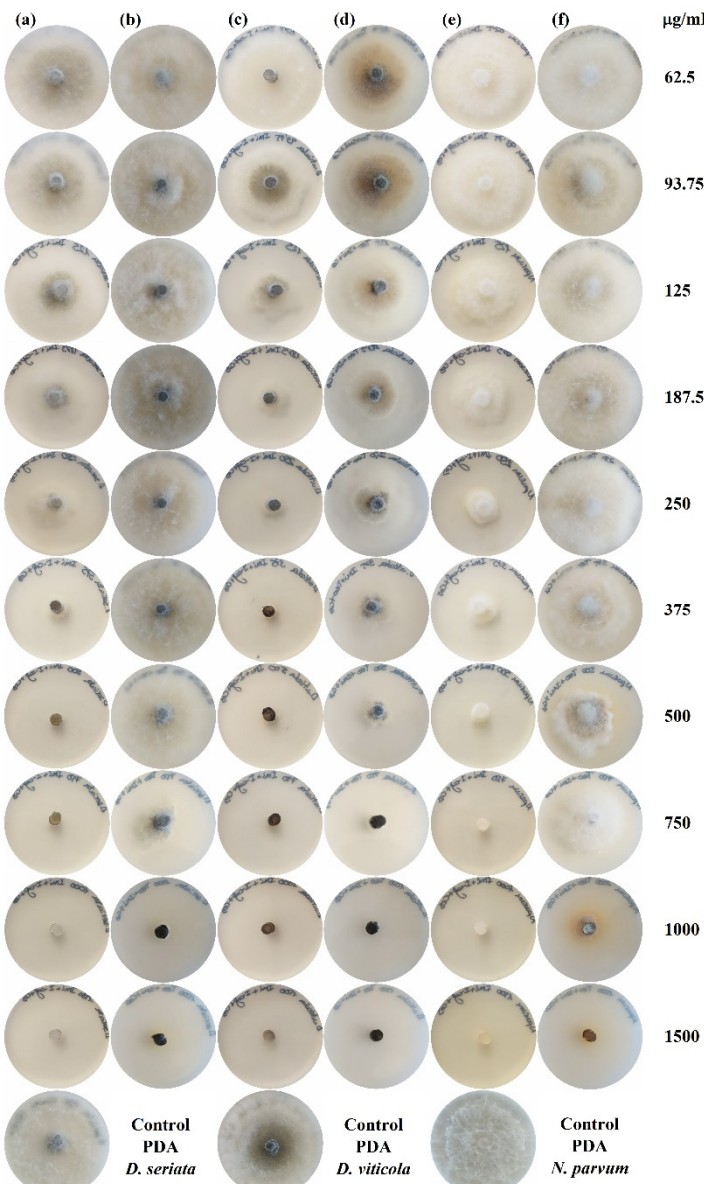

**Figure 7.** Mycelial growth inhibition for the three *Botryosphaeriaceae* fungi (**a,b**) *D. seriata*, (**c,d**) *D. viticola*, and (**e,f**) *N. parvum* upon treatment with COS–I-caffeine-imidazole (**a,c,e**) and COS–theophylline-imidazole (**b,d,f**) ternary conjugate complexes. Only one replicate per treatment and concentration is shown. The control plates (PDA with no treatment) are shown in the bottom row.

**Table 8.** Treatment–lowest concentration combinations that achieve full inhibition of the *Botryosphaeriaceae* fungi.

| *D. seriata* | *D. viticola* | *N. parvum* |
|---|---|---|
| COS–1500 | COS–1500 | COS–1500 |
| COS–F-Phenoxycaffeine–1000 | COS–F-Phenoxycaffeine–1500 | COS–F-Phenoxycaffeine–1500 |
| COS–4F-Phenoxycaffeine–1500 | COS–4F-Phenoxycaffeine–1500 | COS–4F-Phenoxycaffeine–1500 |
| COS–Br-Caffeine–750 | COS–Br-Caffeine–750 | COS–Br-Caffeine–1500 |
| COS–I-Caffeine–500 | COS–I-Caffeine–750 | COS–I-Caffeine–1000 |
| COS–Imidazole–1000 | COS–Imidazole–750 | COS–Imidazole–1500 |
| COS–I-Caffeine-Imidazole–500 | COS–I-Caffeine-Imidazole–375 | COS–I-Caffeine-Imidazole–750 |
| COS–Theophylline–1500 | COS–Theophylline–1000 | COS–Theophylline–1500 |
| COS–Theophylline-Imidazole–1000 | COS–Theophylline-Imidazole–750 | COS–Theophylline-Imidazole–1500 |

## 4. Discussion

### 4.1. On the Structure of the Complexes

After a joint evaluation of the available structural information, it appears that the complexes under study are the result of $\pi$-stacking between the imidazole component and the caffeine half of the COS–caffeine derivative aggregate. Modeling of the caffeine-imidazole derivative aggregates suggests that the least stable species was iodo-caffeine-imidazole. Taking into consideration that, according to the thermal analysis, the COS–iodo-caffeine aggregate was the most stable in this study, an intermediate stability may be expected for the most active complex, COS–iodo-caffeine-imidazole. In any case, it is not the stability itself but the cleavage product that determines the activity of this and the other complexes studied. In this regard, it cannot be excluded that the complexes may behave as COS–based 'nanocarriers' in which the active principle to be conveyed would be the caffeine derivative and imidazole mixture.

### 4.2. On the Activity of the Complexes

#### 4.2.1. Comparison with Imidazole as a Fungicide Control

To assess the efficiency of the proposed treatments based on methylxanthines, alone and in combination with COS, imidazole (known under the commercial name Prochloraz) may fulfill the role of a fungicide control. Although currently not registered for use in vineyards, imidazole has been tested under laboratory, nursery, and vineyard conditions against Botryosphaeria dieback, Esca complex, and Eutypa dieback, and is known to block conidial germination of many GTD pathogens, including *D. seriata, Lasiodiplodia theobromae* (Pat.) Griffon & Maubl., *N. parvum*, *Neofusicoccum australe* (Slippers, Crous & M.J. Wingf.) Crous, Slippers & A.J.L. Phillips, and *Neofusicoccum luteum* (Pennycook & Samuels) Crous, Slippers & A.J.L. Phillips [8].

In view of the effective concentration values reported above (Tables 5 and 6), and based on the results of the two-way ANOVA followed by Tukey's HSD test (Table S1), it may be inferred that caffeine, bromo-caffeine, and iodo-caffeine show significantly stronger antifungal activity than imidazole at most concentrations $\geq$375 µg mL$^{-1}$ against the three fungi, even when used alone. Such differences are even more evident for the binary conjugate complexes. Hence, on the basis of these in vitro results, these treatments are more effective than Prochloraz.

#### 4.2.2. Comparison with Reported Antifungal Activities of Methylxanthines against Wood-Degrading Fungi

Even though susceptibility is species- and isolate-dependent, effective concentrations reported in other works focused on the antifungal activity of caffeine against wood-degrading fungi are discussed here for comparison purposes. Arora and Ohlan [20] investigated the antifungal activity of caffeine against ten different wood-rotting fungi (viz. *Chaetomium globosum* Kunze, *Daldinia concentrica* (Bolton) Ces. & De Not., *Daedalea flavida* Lév., *Flavodon flavus* (Klotzsch) Ryvarden, *Fomitopsis palustris* (Berk. & M.A. Curtis) Gilb. & Ryvarden, *Gloeophyllum trabeum* (Pers.) Murrill, *Phanerochaete chrysosporium* Burds., *Phlebia radiata* Fr., *Pycnoporus sanguineus* (L.) Murrill, and *Sporotrichum pulverulentum* Novobr.) and found full inhibition at concentrations in the 0.3–0.5% range (i.e., 3000–5000 µg mL$^{-1}$). Kwaśniewska-Sip et al. [21] reported that effective protection against *Aspergillus niger* Tiegh., *Aspergillus terreus* Thom, *C. globosum*, *Cladosporium herbarum* (Pers.) Link, *Paecilomyces variotii* Bainier, *Penicillium cyclopium* Westling, *Penicillium funiculosum* Thom, and *Trichoderma viride* Pers. could be attained by soaking Scots pine wood with 10,000 µg mL$^{-1}$ of caffeine, while up to 25,000 µg mL$^{-1}$ was required to completely inhibit *Aspergillus versicolor* (Vuill.) Tirab. and *Phoma violacea* (Bertel) Eveleigh. Kobetičová et al. [5] tested the activity of both caffeine and theophylline at a concentration of 2000 µg mL$^{-1}$ against *Coniophora puteana* (Schumach.) P. Karst., *Gloeophyllum sepiarium* (Wulfen) P. Karst., *Serpula lacrymans* (Wulfen) J. Schröt., and *Trametes versicolor* (L.) Lloyd, finding full inhibition of all fungi by caffeine.

Theophylline was similarly effective against all fungi except *C. puteana*. This difference in efficacy was attributed to the fact that theophylline is degraded more rapidly than caffeine.

It is worth noting that the $EC_{90}$ values reported herein for pure caffeine, its derivatives, and especially for the binary and ternary complexes (Tables 5 and 6) are lower than those reported above, so conjugation of methylxanthines with COS appears to be a promising approach for enhancing their activities with applications beyond the control of GTDs (e.g., to improve wood-preservation treatments).

### 4.3. On Possible Mechanisms of Action

Regarding the mode of action of the treatments assayed in this study, caffeine is known to work in a two-fold manner against fungi: firstly, by directly suppressing growth, and secondly, by promoting mycoparasitism of antagonistic species, such as *Trichoderma* spp., up to 1.7-fold [22]. Regarding the first mechanism, Reinke et al. [23] reported that the fungicidal action of caffeine was based on damaging the cytoplasmic membrane of yeast during the early stages of formation. Wang et al. [24] analyzed the antifungal mechanisms of caffeine for *Colletotrichum fructicola* Prihast., L.Cai & K.D.Hyde in vitro, studying its effects on mycelium morphology, the cell wall, and the plasma membrane. They found that caffeine exerted its antifungal effects through damage to cell walls and membranes, ultimately leading to inhibition of growth or pathogen death, and that the activities of methane dicarboxylic aldehyde (MDA) and superoxide dismutase (SOD) in the hyphae were also affected. Concerning theophylline, its mechanism of fungistatic activity would be similar, involving a damaging effect on the fungal cell wall and cell membrane. Singh et al. [25] investigated the antifungal activity of theophylline against *Candida* spp. and mechanistic insights revealed that it induces membrane damage due to enhanced ionic disturbances and depleted ergosterol levels with a concomitant rise in membrane fluidity due to elevated flippase activity. Regarding the differences in toxicity for fungi among the two methylxanthines, this would result from differences in the dipole moment, lipophilicity, and solubility, as noted by Kobetičová et al. [5]. Caffeine is more lipophilic, which correlates with its easier penetration of membranes and possible toxic effects on fungi.

In relation to the mode of inhibition for the other two compounds studied, several mechanisms of action have been proposed for COS [26,27]. The interaction of the positively charged COS with negatively charged phospholipid components would result in increased permeability and the leakage of cellular contents; its chelating action would deprive fungi of trace elements essential for normal growth; and binding to fungal DNA would inhibit mRNA synthesis and affect protein and enzyme production. Regarding the imidazole-based compounds, their mechanisms of antimycotic action would include arresting nuclear division, binding to tubulin, and inhibiting microtubule assembly [28].

With respect to the mechanism for the synergism observed for the binary and ternary conjugate complexes, the possible stimulation of multiple functions has been advocated when a synergistic behavior has been observed for caffeine with other antifungal products [29]. According to Tayel et al. [30], the synergism between chitosan and plant extracts can be tentatively explained by the various fungicidal components of each agent and by the fact that fungal pathogens are not readily resistant to multiple fungitoxicants. The same hypothesis may be also applicable to chitosan-imidazole binary combinations (without caffeine in their composition), for which Sabaa et al. [31] also reported antimicrobial activity, and to the ternary conjugate complexes.

### 4.4. Limitations of the Study and Future Research Lines

Several limitations of this study should be brought to the reader's attention. Firstly, the study presented herein fulfills a screening purpose, designed to shortlist—based on in vitro results—those treatments that may be the most effective for the control of GTDs. Nonetheless, in vivo and field (multi-year and multi-location) studies will be needed to confirm the real-world applicability of these treatments, define appropriate dosages, and

determine the most suitable application procedure (e.g., via fertigation or endotherapy using nanocarriers [32]).

Secondly, phytotoxicity studies should also be conducted. While it is well-established that benzimidazoles (such as Prochloraz) may have some phytotoxic effects (namely, they can decrease plant biomass and induce a considerable reduction in chlorophyll a, chlorophyll b, carotenoids, and total pigments content [33]), and that chitosan oligomers are safe (phytotoxic effects have been discarded for chitosan-based treatments by in vivo assays with grapevine plants [34]), no data on the phytotoxicity of methylxanthines in grapevine plants are available in the literature. In fact, there are conflicting reports on the phytotoxic effect of methylxanthines in other plants: while some authors have reported that caffeine, theobromine, and theophylline inhibited the germination of lettuce at 400 ppm [35], other authors have reported that caffeine (0.2%) can be safely used for the control of potato blackleg caused by *Dickeya solani* [36].

Thirdly, an in-depth study of the underlying mechanism for the synergism observed for the binary and ternary combinations would be a particularly important topic for further study, since it may help pave the way for the design of more effective antifungals for GTDs.

## 5. Conclusions

Caffeine, four of its derivatives, and one of its natural metabolites (theophylline) were investigated, alone and after conjugation with chitosan oligomers, for the control of three pathogenic fungi associated with grapevine trunk diseases. In vitro antifungal susceptibility tests determined that the COS–iodo-caffeine complex had the best overall antifungal activity against the three *Botryosphaeriaceae* species tested, with an $EC_{90}$ of 471, 640, and 935 μg mL$^{-1}$ against *D. seriata*, *D. viticola*, and *N. parvum*, respectively, performing significantly better than a conventional fungicide (imidazole, i.e., Prochloraz), which registered corresponding $EC_{90}$ values of 2793, 1190, and 2392 μg mL$^{-1}$, respectively. Given that benzimidazoles are among the most effective active substances for GTDs among those tested in the literature, the formation of ternary complexes with imidazole was also explored. This strategy identified a synergistic effect, resulting in $EC_{90}$ values in the range 271–509 μg mL$^{-1}$ for the COS–iodo-caffeine-imidazole complex. Although in vivo and field studies are still needed to confirm the real-world applicability and safety of these treatments, the in vitro results reported here suggest that COS–methylxanthine-based conjugate complexes hold promise for the control of GTDs and may also contribute to enhancing the activity (and thus help reduce the concentrations needed) of conventional imidazolic fungicides (e.g., benomyl, carbendazim, methyl thiophanate, etc.).

**Supplementary Materials:** The following supporting information can be downloaded at: https://www.mdpi.com/article/10.3390/agronomy12040885/s1, Figure S1: Mycelial growth inhibition of *D. seriata* upon treatment with COS, caffeine, F-phenoxycaffeine, 4F-phenoxycaffeine, Br-caffeine, I-caffeine, imidazole, and theophylline at different concentrations; Figure S2: Mycelial growth inhibition of *D. viticola* upon treatment with COS, caffeine, F-phenoxycaffeine, 4F-phenoxycaffeine, Br-caffeine, I-caffeine, imidazole, and theophylline at different concentrations; Figure S3: Mycelial growth inhibition of *N. parvum* upon treatment with COS, caffeine, F-phenoxycaffeine, 4F-phenoxycaffeine, Br-caffeine, I-caffeine, imidazole, and theophylline at different concentrations; Figure S4: Mycelial growth inhibition of *D. seriata* upon treatment with COS conjugated complexes. COS–caffeine, COS–F-phenoxycaffeine, COS–4F-phenoxycaffeine, COS–Br-caffeine, COS–I-caffeine, COS–imidazole, and COS–theophylline at different concentrations; Figure S5: Mycelial growth inhibition of *D. viticola* upon treatment with COS conjugated complexes. COS–caffeine, COS–F-phenoxycaffeine, COS–4F-phenoxycaffeine, COS–Br-caffeine, COS–I-caffeine, COS–imidazole, and COS–theophylline at different concentrations; Figure S6: Mycelial growth inhibition of *N. parvum* upon treatment with COS conjugated complexes. COS–caffeine, COS–F-phenoxycaffeine, COS–4F-phenoxycaffeine, COS–Br-caffeine, COS–I-caffeine, COS–imidazole, and COS–theophylline at different concentrations; Table S1: Summary of multiple pairwise comparisons using Tukey's HSD test for each treatment–concentration combination against each of the three phytopathogenic fungi.

**Author Contributions:** Conceptualization, C.A.-J. and J.M.-G.; methodology, C.A.-J., L.B.-D. and A.C.-G.; validation, C.A.-J. and A.C.-G.; formal analysis, E.S.-H., L.B.-D. and P.M.-R.; investigation, E.S.-H., C.A.-J., L.B.-D., A.C.-G., J.M.-G. and P.M.-R.; resources, C.A.-J. and J.M.-G.; writing—original draft preparation, E.S.-H., C.A.-J., L.B.-D., A.C.-G., J.M.-G. and P.M.-R.; writing—review and editing, E.S.-H., J.M.-G. and P.M.-R.; visualization, E.S.-H.; supervision, P.M.-R.; project administration, J.M.-G.; funding acquisition, J.M.-G. All authors have read and agreed to the published version of the manuscript.

**Funding:** This research was funded by Junta de Castilla y León under project VA258P18 with FEDER co-funding.

**Institutional Review Board Statement:** Not applicable.

**Informed Consent Statement:** Not applicable.

**Data Availability Statement:** The data presented in this study are available on request from the corresponding author. The data are not publicly available due to their relevance to an ongoing PhD thesis.

**Conflicts of Interest:** The authors declare no conflict of interest. The funder had no role in the design of the study; in the collection, analyses, or interpretation of data; in the writing of the manuscript, or in the decision to publish the results.

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
