# Peer review of "Antifungal Activity of Methylxanthines against Grapevine Trunk Diseases"

_agronomy, doi:10.3390/agronomy12040885_

Round 1

Reviewer 1 Report

The manuscript entitled "Antifungal Activity of Caffeine Derivatives against Grapevine Wood Diseases" may be considered for publication after the revision of the following:
The purpose of this study is not mentioned in the abstract. There are no results (short term) and no conclusion.
The introduction is poor. The purpose of this study is not mentioned here either. I recommend improving the input with data that can be correlated with the results in this manuscript.
I did not see the statistical analysis in the tables.
The discussions are brief. Very few correlations and extensive discussions with existing studies in the literature.

Reviewer 2 Report

Grapevine trunk diseases (GTDs) has become a major challenge faced by modern viticulture. It is valuable to develop suitable fungicide to control GTDs, as there is still no high effective fungicide to this diseas up to now.However I have some concerns for the authors about the other parts:

  1. More information should be included in the introduction part, as the authors did not provide enough information on why the authors want to do this research. And refer the reason why do research on theophylline, the authours didnot introduce it.
  2. The objective of develop fungicide is to improve the control efficiency, so I want to know if the authors set the fungicide control when conduct the in vitro antimicrobial activity assessment. I recommend the authors including this informatin.
  3. When refer the pathogens' latin name for the first time, the full name should be used. While refer to the latin name again in the same manuscript, the abbreviated names should be used.  Abbreviations of the latin names have certain rules, the author needs to find professional assistance to verify confirmation.
  4. Line 353 the citatioin of the second mode of action of caffeine is not quite appropriate. It should be assisting the mycoparasitism of the pathogens' natural enemy.

Reviewer 3 Report

The manuscript entitled “Antifungal Activity of Caffeine Derivatives against Grapevine Wood Diseases” is quite interesting and requires some corrections/ clarifications before publishing which is detailed below….

  • Abstract “These results suggest that caffeine derivatives conjugated with COS, with or without imidazole, may be promising treatments for the control of GTDs”. I don’t agree with the statement has the authors have not done in vivo evaluation
  • The MS lacks citation of recent references.
  • For antifungal assay, photographs should be given and the bar graphs can be represented in table form.
  • Why the authors have not gone for Two-way ANNOVA? Which I found is suitable to identify the best concentrations and best compound against the CTDs.
  • Line 136: concentrations in the 62.5−1500 μg·mL-1 range; if there is any toxicity to the plants by the synthesized compounds should be provided?
  • Line 142: why the antifungal activity studies were conducted in dark conditions? Justify?
  • Unit of expression in Table 5 is missing
  • Line 355-356: “the mechanism behind the synergism observed for the combination of caffeine with imidazole or COS and, of course, that of caffeine derivatives with imidazole and COS” should be carried out in order to authenticate its efficacy in managing the GTDs
  • Line 379: “an environmentally friendly alternative to conventional fungicides” has not been proved in the study? and should be rewritten.
  • The authors have provided the effective concentration against each of the pathogen, but they have not mentioned among the different compounds evaluated, which is the best for all the three CTDs.
  • The manuscript lacks the toxicity analysis before claiming the compounds as possible alternatives for the management of GTDs.
  • Discussion needs further revision.
  • Conclusion is not conclusive and it is written in general and hence should be modified completely on emphasizing the results obtained during the study.

Round 2

Reviewer 1 Report

The authors responded to all my comments.

Reviewer 3 Report

Accept